# Recommendations to Synthetize Old and New β-Lactamases Inhibitors: A Review to Encourage Further Production

**DOI:** 10.3390/ph15030384

**Published:** 2022-03-21

**Authors:** Silvana Alfei, Guendalina Zuccari

**Affiliations:** Department of Pharmacy (DIFAR), University of Genoa, Viale Cembrano, 16148 Genoa, Italy; zuccari@difar.unige.it

**Keywords:** optimized synthetic procedures, β-lactam antibiotics, β-lactamase enzymes, β-lactamase enzymes inhibitors, multi-drug resistant (MDR) bacteria, serine β-lactamases, metal-β-lactamases

## Abstract

The increasing emergence of bacteria producing β-lactamases enzymes (BLEs), able to inactivate the available β-lactam antibiotics (BLAs), causing the hydrolytic opening of their β-lactam ring, is one of the global major warnings. According to Ambler classification, BLEs are grouped in serine-BLEs (SBLEs) of class A, C, and D, and metal-BLEs (MBLEs) of class B. A current strategy to restore no longer functioning BLAs consists of associating them to β-lactamase enzymes inhibitors (BLEsIs), which, interacting with BLEs, prevent them hydrolyzing to the associated antibiotic. Worryingly, the inhibitors that are clinically approved are very few and inhibit only most of class A and C SBLEs, leaving several class D and all MBLEs of class B untouched. Numerous non-clinically approved new molecules are in development, which have shown broad and ultra-broad spectrum of action, some of them also being active on the New Delhi metal-β-lactamase-1 (NDM-1), which can hydrolyze all available BLAs except for aztreonam. To not duplicate the existing review concerning this topic, we have herein examined BLEsIs by a chemistry approach. To this end, we have reviewed both the long-established synthesis adopted to prepare the old BLEsIs, those proposed to achieve the BLEsIs that are newly approved, and those recently reported to prepare the most relevant molecules yet in development, which have shown high potency, providing for each synthesis the related reaction scheme.

## 1. Introduction

The chemical structure of β-lactam antibiotics (BLAs) used to treat a large variety of infections [1] comprises a bicyclic or monocyclic structure. In both cases, a four-terms β-lactam nucleus is present, which embodies the weak point of these drugs, being the target of β-lactamase enzymes (BLEs). BLEs are enzymes produced by an increasing number of clinically relevant bacteria as defensive strategy against BLAs. Specifically, they catalyse the hydrolysis and inactivating opening of the four-membered β-lactam cycle [2].

The most established strategy to prevent the hydrolytic activity of BLEs consists of developing BLEs inhibitors (BLEsIs), generally free of intrinsic antibacterial effects. These molecules, when administered in associations with already existing BLAs no longer functioning, can protect them, thus allowing them to recover their original activity.

The currently available classes of BLAs derive from structural modifications of the natural penicillin, executed to prepare semisynthetic antibiotics with increasingly enhanced characteristics, and capable of circumventing the emergence of bacterial resistance [1]. Bicyclic nuclei are present in penicillin-like BLAs, cephalosporins, and carbapenems, while monocyclic systems are typical of monobactams. Particularly, penicillin and derivatives shown in Figure 1a,b have a four-membered β-lactam ring fused to a five-membered ring [1,2]. Compound (**1**) shows the correct numbering of the basic system, which can be applied also to compounds (**2**), (**3**), (**4**), (**5**), and (**6**). Cephalosporins and derivatives are characterized by a six-membered ring fused with the β-lactam nucleus (Figure 1c,d), while a monocyclic nucleus and a sulphonic acid group on the nitrogen atom of the cyclic amide typifies monobactams (Figure 1e) [1,2]. Compound (**7**) shows the correct numbering of the basic system, which can be applied also to compounds (**8**), (**9**), (**10**), (**11**), and (**12**), while the correct numbering of the monobactam system has been shown on compound (**13**).

More generically, penicillins having a thiazolidine five membered ring are named penams. Logically, penicillin derivatives in which the sulfur atom has been substituted by an oxygen atom are named oxapenams, and those having a carbon atom in place of the sulfur one are named carbapenams (Figure 1a, compounds (**1**), (**2**), and (**3**)). The derivatives of penams, carbapenams, and oxapenams having double bonds (C=C) in position 2–3 in the five-terms rings create the families of penems, carbapenems, and oxapenems (Figure 1b, compounds (**4**), (**5**), and (**6**)). Cephalosporins, which have double bonds in position 2–3 in the six-terms ring, are named cephems, therefore their derivatives obtainable by changing the sulfur atom with a carbon atom or with an oxygen one generate the families of carbacephem and oxacephem, respectively (Figure 1c, compounds (**7**), (**8**), and (**9**)). Finally, the correspondent saturated derivatives engender the families of cepham, carbacepham, and oxacepham (Figure 1d, compounds (**10**), (**11**), and (**12**) [3].

The mechanism of action of BLAs involves their deactivating interaction with a set of D-alanyl-D-alanine transpeptidases enzymes, identified as penicillin binding proteins (PBPs), which catalyse the synthesis of the peptidoglycan layer of the bacterial cell wall [4]. Without functioning PBPs, the synthesis of peptidoglycan cannot occur, bacteria cannot grow, and consequently they die [5]. While bicyclic BLAs, such as penicillin and cephalosporin, have a broad spectrum of action, which includes both Gram-positive and Gram-negative bacteria, the monobactams are specifically active against aerobic isolates of Gram-negative species [6,7]. Importantly, carbapenems are effective against both Gram-positive and Gram-negative, also including multidrug resistant (MDR) bacteria. Nowadays, carbapenems constitute an essential group of antibiotics for the treatment of severe infections caused by frightening super-bugs, resistant to the most part of traditional antibiotics [8]. Structurally, as above mentioned, carbapenems are bicyclic β-lactam molecules like penicillin, but having a carbon atom in position 4 in place of the sulphur atom of penicillin and containing a C2-C3 double bond, they are more resistant to the hydrolytic attack of the most part of BLEs (Figure 2) [5].

Concerning the available antibacterial armamentarium, in terms of β-lactam antibiotics, the main types of BLAs developed until now have been reported in Table 1, constructed using the public information provided online at https://en.wikipedia.org/wiki/List_of_%CE%B2-lactam_antibiotics# (accessed on 10 February 2022) [3]. 

Unfortunately, to challenge the dangerous action of BLAs, bacteria are increasingly developing resistance by different mechanisms, including the hyper-expression of efflux pumps, the down regulation of the expression of porins, the alteration of PBPs and, above all, the production of β-lactamase enzymes (BLEs), which inactivate BLAs [9,10].

Particularly, BLEs can hydrolyse the β-lactam amide bond present in the four-terms cycle of BLAs. This process causes the ring opening and other irreversible molecular modifications, thus leading to ineffective products, because they are no longer capable of interacting with PBPs [5]. Based on the structural features of their active-site, BLEs have been classified in serine BLEs (SBLEs) [11] and in metallo-BLEs (MBLEs). The latter ones possess a very broad spectrum of action, and nowadays, although particularly in Asia, they are disseminated worldwide [12,13]. According to the Ambler classification, SBLEs have been grouped into class A, C, and D, while MBLEs give rise to the BLEs of class B [1,14,15,16]. 

Initially, SBLEs were capable of hydrolysing only the first generation BLAs, such as narrow and broad-spectrum penicillin and penicillin derivatives. Therefore, the development of extended spectrum cephalosporins of different generations (cefotaxime, ceftriaxone, and ceftazidime) allowed to counteract severe infections by bacteria producing such enzymes. In a second moment, the emergence of extended spectrum SBLEs (ESSBLEs), also capable of hydrolysing the β-lactam ring of cephalosporines, dramatically reduced the therapeutic potential of the available antibacterial armamentarium. In this contest, the development of carbapenems, which displayed a wider spectrum of action, and were capable to resist the hydrolytic action of several ESSBLEs, were also effective against alarming bacteria producing ESSBLEs, and their clinical application strongly increased in the 2000s [17]. Carbapenems, including the longest established imipenem, meropenem, and ertapenem, and the recently developed doripenem, biapenem, panipenem, razupenem, and tomopenem [17], in addition to resist the inactivating action of the first generations of ESSBLEs, demonstrated their ability to also act as inhibitors for many ESSBLEs [18,19,20]. However, the excessive and inappropriate continued use of BLAs has selected, in the past, decade bacteria that is also resistant to carbapenems, including doripenem, ertapenem, imipenem, and meropenem [8,21,22], due to their capability to produce both new ESSBLEs capable to also hydrolyse carbapenems and carbapenem-hydrolysing MBLEs of group B [23,24,25]. Additionally, the reduction of membrane porins, the improvement in number of efflux pumps [23], the alteration of PBPs (PBP2 or PBP3), and the hyperexpression of class C AmpCs are supplementary processes involved in the resistance of bacteria to carbapenems [26,27]. Specifically, the global worry mainly regards the proliferation of certain types of bacteria resistant to carbapenems, which includes those capable of producing the *Klebsiella pneumoniae* carbapenemases (KPCs) belonging to the ESSBLEs of class A, the oxacillinases (OXAs) belonging to the ESSBLEs of class D, and above all the New Delhi metallo-β-lactamases (NDMs and mostly NDM-1) belonging to MBLEs of class B [1,16]. Notably, NDMs are a subclass of recently identified enzymes which makes bacteria resistant to a broad spectrum of BLAs, including carbapenems family, which have been a staple for the treatment of antibiotic-resistant bacteria.

In the list published by the WHO in the year 2017, carbapenem-resistant *Acinetobacter baumannii* (CRAB), carbapenems-resistant *Pseudomonas aeruginosa* (CRPA), as well as carbapenem- and third generation cephalosporin-resistant *Enterobacteriaceae* (CRE), were inserted in the highest category of urgency [28]. The long-established strategy to prevent the resistance to BLAs, due to the increasing diffusion of bacteria producing BLEs, consists of developing BLEs inhibitors (BLEsIs) to administer in associations with already existing BLAs for preventing their inactivation. The discovery and subsequent commercialization of first generation β-lactam-type BLEsIs, such as clavulanic acid, sulbactam, and tazobactam, strongly ameliorated the alarming scenario of the antimicrobial therapy. Despite being free of any antibacterial effect, when co-administered with no longer functioning BLAs, such BLEsIs allowed them to recover their original activity. Combinations including amoxicillin/clavulanic acid, ampicillin/sulbactam, and piperacillin/tazobactam have been successfully used over a long period and are currently in use [16,22]. Unfortunately, the activity of these inhibitors is limited mainly to the BLEs of class A, not including carbapenemases, and is weak against class C BLEs and absent against carbapenemases of group B and D. Consequently, due to the increasing emergence and diffusion of serine carbapenemases of class A and D, such as KPCs and OXAs, respectively, and carbapenemases of class B, such as NDMs, new β-lactam and non-β-lactam inhibitors including the diazobicyclooctanes (DBOs) and the cyclic boronic acid derivatives have emerged and are in development [1,29]. Among several molecules synthetized and under investigations, only avibactam and relebactam among DBOs and vaborbactam among the boronic acid derivative have been approved and are currently clinically applied [14,30]. However, the commercialization of these new inhibitors does not address the need for new compounds able to inactivate the increasing number of KPCs and OXAs-like carbapenemases and the ultra-broad spectrum carbapenemases of class B, including the NDM-1, which can hydrolyse almost all BLAs except aztreonam [31]. Currently, the inhibition of NDM-1 is the main goal pursued by researchers and a variety of NDM-1 inhibitors have been found, including captopril, thiol compounds, Aspergillomarasmine A, new DBO and boric acid derivatives, sulphanilamide compounds, succinic acid derivatives, and a series of natural products. Unfortunately, their physicochemical properties, specificity, and safety to the human body still hamper the approval of their clinical use, and no anti-NDM-1 inhibitor has been marketed until now [31]. The scope of this review was to provide the state of the art concerning the BLEsIs developed so far, including both those already clinically approved and marketed and those at the various stages of development. Originally, to not duplicate the existing reviews concerning this topic, in this work, we have examined BLEsIs by a chemistry approach, with particular attention to their chemical structure. To this end, we have herein reviewed the long-established synthesis adopted to prepare the old BLEsIs, those reported to afford the BLEsIs recently approved, and those lately described to prepare the most relevant molecules yet in development, which have shown potent inhibiting activity. To make the understanding of the described synthetic procedures easy, for each one, the relative synthetic scheme has been also supplied.

## 2. β-Lactamase Enzymes (BLEs)

Before delving into the world of BLEsIs, we felt it was our duty to provide some essential information on the most clinically relevant BLEs. BLEs are produced by both Gram-positive and Gram-negative bacteria to develop resistance to BLAs. Such bacteria include methicillin resistant *Staphylococcus aureus* (MRSA), *Enterobacteriaceae* such as *K. pneumoniae*, *Citrobacter* supp, *Proteus vulgaris*, *Morganella* supp, *Salmonella* supp, *Shigella* supp, *Escherichia coli*, *Haemophilus influenzae*, *Neisseria gonorrhoeae*, as well as by *Pseudomonas aeruginosa*, *Acinetobacter baumannii*, and *Mycobacterium tuberculosis* [22]. Genes encoding BLEs can be either on bacterial chromosomes and are not horizontally transmittable, or on horizontally transmittable plasmids. According to the primary sequence homology and Ambler classification, BLEs have been arranged into four classes, namely A, B, C, and D [32]. BLEs of class A, C, and D belong to the SBLEs, which catalyse the hydrolysis of BLAs via a serine-bound acyl ester intermediate [11]. Such intermediate is formed thanks to a nucleophilic acyl substitution reaction, due to the attaching of the hydroxyl group of the serine present in the active site of the enzyme to the carbonyl group of the β-lactam ring of BLAs, thus triggering its opening. Next, the nucleophilic attack by a water molecule cleaves the covalent bond between the enzyme and the carbonyl group of the β-lactam ring, generating a carboxylic group. This allows the degraded β-lactam antibiotic to diffuse away, thus freeing up the enzyme to process other β-lactam molecules. Additionally, a decarboxylation can occur, thus leading to the complete and irreversible inactivation of BLAs (Figure 1). 

Differently, the class B enzymes belong to the MBLEs, which require zinc for activity. Particularly, MBLEs cleave β-lactams by a mechanism like that of metalloproteases, which are protease enzymes whose catalytic mechanism involves a metal. No covalent intermediate is formed, and this is the reason of the ineffectiveness of the currently clinically approved BLEsIs, which work binding BLEs covalently. Regarding this, the spread of bacterial strains expressing MBLEs, such as the NDM-1, has engendered considerable concerns [33]. 

The metal ion zinc (Zn^2+^), which catalyses the hydrolytic action of MBLEs, is coordinated to the protein via three ligands, which can vary with histidine, glutamate, aspartate, lysine, and arginine. The fourth coordination position is taken up by a labile hydrolysing water molecule, which is activated by the divalent cation. Treatment with chelating agents, such as EDTA, leads to the complete inactivation of the enzyme, due to its chelator capability that removes zinc.

The three-dimensional (3D) crystal structures of several class A, C, and D BLEs and those of bacterial transpeptidases, including PBPs, are similar, thus suggesting a common evolutionary origin. On the contrary, the 3D structures of several class B enzymes have a lack of similarity with those of the SBLEs and transpeptidases, thus indicating a different and independent evolutionary origin. 

Collectively, more than 400 different types of BLEs have been reported. Table 2 provides a list of the most clinically relevant enzymes belonging to the class A, B, C, and D. 

TEMs are the most common class A BLEs, whose genes are present in plasmids of Gram-negative bacteria. At least 140 BLEs of TEM-type have been currently reported. The ESSBLEs of class A named SHVs, of which 60 varieties are known, are spread worldwide, but are predominant in Europe and in the United States [35,36,37]. 

More than 40 class A ESSBLEs of CTX-M-type are known, including CTX- M-14, CTX-M-3, and CTX-M-2, and are particularly spread in South America, prevailing over TEM and SHV during the first decade of the 21st century [35,36,37,38].

Class A carbapenemases of KPC family are encoded in plasmids and are the most widely spread enzymes hydrolysing carbapenems distributed worldwide [34].

Several *Enterobacteriaceae* and a few other bacteria expressing the chromosomal AmpC gene produce the class C AmpC ESSBLEs, which can inactivate several penicillin derivatives and cephalosporins and many combinations BLEsIs/BLAs. Additionally, in *E. coli*, *K. pneumoniae*, and *P. mirabilis*, which are missing the chromosomal AmpC gene or poorly express it, a transmissible plasmid encoding gene for AmpC enzymes can appear [35,36,37]. 

Antibiotics of the carbapenems family are usually effective to treat infections by bacteria producing AmpC ESSBLEs, but carbapenem resistance has been observed in some pathogens due to other phenomena associated with the overexpression of the AmpC gene, such as the reduction of outer membrane porins, or the enhancement of the BLAs efflux by an increased activation of efflux pumps [35,36,37].

Generally, other class C BLEs do not provide resistance to carbapenems, but five enzymes belonging to this group recently identified and named ACT-1, DHA-1, CMY-2, CMY-10, and ADC-68 have been shown to be capable of hydrolysing carbapenems [39].

The OXA-type BLEs belonging to the class D, except for OXA-18, are carried on plasmids. Bacteria producing OXAs are resistant to several BLAs, including carbapenems, but except for OXA-48-producers *Enterobacteriaceae*, OXAs producing bacteria go back to being susceptible if clavulanic acid is associated. Interestingly, although the discovery of class D OXA-type carbapenemases (also named carbapenem hydrolysing class D lactamase (CHDLs) dates to many years ago, their rapid spread is very recent [1,40].

Other enzymes, such as *Pseudomonas* extended resistance (PERs), Vietnam extended-spectrum (VEBs), Serratia fonticola (SFO-1), Tlahuicas (Indian tribe) (TLA-1), Brazilian ESSBLEs (BES-1), GESs Belgium ESSBLEs (BEL-1), and TLA-2, which has 51% amino-acid identity with TLA-1, are uncommon plasmids-mediated ESSBLEs, which are produced mainly by *P. aeruginosa* [35,36,37].

The MBLEs of class B are enzymes mediated by plasmids that, although initially found only in poorly pathogenic environmental bacterial species, are now found in several clinically relevant strains of Gram-negative bacteria [42]. According to similarities in the sequences, MBLEs are grouped into subclasses B1, B2, and B3 [43]. Among MBLEs, IMPs, VIMs, GIMs, SIMs, SPMs, and NDMs represent the most common families [42]. The first transferable MBLE was IMP-1, which was found in an isolate of *P. aeruginosa* in Japan in 1988, and now is detected worldwide into the *Enterobacteriaceae*. Indeed, IMPs were reported in Europe (1997), Canada, and Brazil. VIM-1 (Verona integron-encoded metallo-β-lactamase-1), comprising 14 varieties of enzymes now globally distributed, was found in Verona, Italy, in 1997. While rare in *Enterobacteriaceae*, VIM-type ESSBLEs are commonly produced by *P. aeruginosa* and *P. putida.*

SPM-1 (Sao Paulo MBLE), GIM-1 (German imipenemase) and SIM-1 (Seoul imipenemase) were isolated in Sao Paulo (Brazil) in 1997, in Germany (2002), and in Korea (2003), respectively.

Worryingly, bacteria carrying the NDM-1 resistance gene, which was first isolated in India and has now spread worldwide, are considered a new type of “superbug” able to inactivate almost all BLAs, except aztreonam, thus leading to the development of multidrug-resistant bacteria [44].

Particularly, NDM-1 belongs to the B1 subclass of MBLEs of group B and is composed of a single polypeptide chain of 27.5 KDa. Three loop regions, L3, L7, and L10, form the main part of the active site, which allocates at the bottom two positively charged zinc ions (Zn1/Zn2) having catalytic activity [45,46,47]. The broad-spectrum activity of NDM-1 is due to the presence of a wide and shallow substrate-binding capsule cavity, which is capable of easily accepting different types of BLAs.

## 3. Old and Novel β-Lactamase Enzymes Inhibitors (BLEsIs)

Except for some molecules recently developed, BLEsIs are compounds without intrinsic antibacterial effects. Administered in association with BLAs no longer effective because they were hydrolyzed by BLEs, their main objective is to protect the associated antibiotics. Particularly, by binding the active site of the BLEs, BLEsIs prevent their hydrolytic action on BLAs, which recover their original activity [48,49]. 

Currently, the available armamentarium of inhibitors and of associations BLAs/BLEsIs clinically approved and already used in therapy to treat infections by bacteria producing BLEs is very poor, and their spectrum of action is still too limited. Presently, the available marketed BLEsIs are only six, of which tazobactam, clavulanate, and sulbactam are effective mainly against SBLEs and ESSBLEs of class A, not including carbapenemases, and avibactam is also active on class A carbapenemases of KPC group, AmpC enzymes of class C, and OXA-48 carbapenemases of class D, while relebactam and vaborbactam possess a spectrum of action slightly narrower than that of avibactam since they are not active against OXA-like enzymes [1,30]. No already approved BLEsI is active against MBLEs. Mainly, BLEsIs are classified as β-lactam inhibitors having a β-lactam core, and as non-β-lactam inhibitors having either a DBO core or other types of non-β-lactam cores. Among the clinically approved inhibitors, clavulanic acid or clavulanate, sulbactam, tazobactam, and tebipenem pivoxil (approved and marketed only in Japan) are β-lactam inhibitors, avibactam and relebactam are non-β-lactam inhibitors having a DBO core, and vaborbactam is a non-β-lactam inhibitor having a cyclic boronic acid core [50,51,52,53]. Interestingly, while β-lactam inhibitors form an irreversible bond with the active site of BLEs, avibactam and the other non-β-lactam inhibitors not containing a β-lactam ring bind reversibly to BLEs [48,54].

In addition to the currently marketed BLEsIs, several new molecules that are not yet on the market, which are still at preclinical studies or even at very early-stages of development, have been synthetised. Table 3 report the structures of the most relevant BLEsIs already clinically approved and marketed, as well as those of bioactive molecules not yet approved and only at preclinical trials or even at the in vitro experimentation step. Considering the most clinically relevant BLEs, the spectrum of action of BLEsIs reported in Table 3 has been also evidenced in an eye-catching way.

## 4. BLAs/BLEsIs Combinations Currently Available on the Market and Clinically Applied

The following Table 4 summarizes the clinically approved BLAs/BLEsIs combinations currently marketed in Europe and USA, with the most common brand names, the available dosage forms, as well as the suggested clinical uses and dosage for both adult patients and paediatric ones.

## 5. Marketed and Not Marketed BLEsIs: Guidelines for Their Synthesis 

### 5.1. β-Lactams 

Generally, the synthetic procedures reported to prepare β-Lactams BLEsIs start from the naturally occurring (+)-6-aminopenicillanic acid (6-APA). Clavulanic acid is instead prepared biosynthetically, as reported in the subsequent Section, while the synthesis of tebipenem pivoxil has been reported starting from a commercially available 6-APA derivative. 

#### 5.1.1. Already Clinically Approved β-Lactams BLEsIs

##### Biosynthesis of Clavulanic Acid

Clavulanic acid (CA) was first identified from *Streptomyces clavuligerus* and later from other *Streptomyces* species. Structurally, clavulanic acid is characterized by a bicyclic nucleus comprising a four-membered β-lactam ring fused to a five-membered oxazolidine ring. While resembling penicillin, CA has an oxygen atom in place of a sulfur one in the five-membered cycle, does not possess an acylamino substituent, and has an exo-β-hydroxy ethylidene group attached at C-2. Nowadays, CA is produced by the biosynthetic pathway in Figure 2 [143].

While arginine and ornithine were proposed to be possible precursors providing the nitrogen atom to obtain CA, in 1993 it was demonstrated that arginine is a more direct precursor than ornithine. Additionally, it was assessed that the β-lactam ring of CA derives from glyceraldehyde-3-phosphate, and that carboxyethyl arginine synthase (CEAS), having cofactors Mg^2+^ and thiamine diphosphate (ThDP) present in the active site, is the enzyme responsible for combining these two precursors, providing the N^2^-carboxyethyl-arginine. β-lactam synthetase (BLS) catalyzes the second step of the pathway converting the N^2^-(2-carboxyethyl) arginine to deoxy-guanidino-proclavaminic acid, thereby generating the monocyclic β-lactam ring. The catalytic reaction of BLS proceeds by adenylation of the N^2^-(2-carboxyethyl) arginine β-carboxylate followed by cyclization via an oxo-anion intermediate in the presence of ATP and Mg^2+^. Two other enzymes, namely clavaminate synthase (CAS) and proclavaminate amidino hydrolase (PAH), catalyze the subsequent reactions to generate the bicyclic intermediate, namely clavaminic acid. Particularly, deoxy guanidino proclavaminic acid is first hydroxylated by CAS to give guanidino proclavaminic acid, which under the action of PAH that removes the arginine-derived guanidino group, is converted to proclavaminic acid. Then, CAS contributes again to the next two reactions, initially catalyzing the formation of the first bicyclic intermediate through oxidative ring closure of proclavaminic acid to give dihydroclavaminic acid, and then forming clavaminic acid by desaturation. Then, N-glycyl-clavaminic acid synthetase (GCAS), which is a member of the ATP-grasp fold superfamily, catalyzes the conversion of clavaminic acid to N-glycyl-clavaminic acid in an ATP-dependent manner. Although the mechanism by which N-glycyl-clavaminic acid is converted to clavaldehyde is still not clear, it is assumed that a double epimerization and an oxidative deamination reaction should occur to ultimately yield clavaldehyde. Clavaldehyde, otherwise 3R,5R-clavulanate-9-aldehyde, is an unstable α, β-unsaturated aldehyde with β-lactamase inhibitory activity, which finally undergoes reduction to give clavulanic acid in an NADPH-dependent reaction catalyzed without degradation of the bicyclic β-lactam ring by clavulanic acid dehydrogenase (CAD).

##### Synthesis of Sulbactam

According to the patent by Chen et al. [144], sulbactam acid was prepared according to Figure 3. Briefly, bromine, a diluted sulfuric acid solution, and solid sodium nitrite were added to an aqueous solution of 6-APA **1** to perform diazotization, achieving the intermediate **2** and bromination to obtain intermediate **3**. Then, an aqueous solution of potassium permanganate and diluted sulfuric acid was added dropwise to the intermediate **3** to perform an oxidation reaction and obtain the sulphone derivative **4**, which was treated with strontium powder and diluted sulfuric acid to perform its reductive debromination and obtain sulbactam in a yield of 90%. 

##### Synthesis of Tazobactam

Different synthetic procedures have been reported in the literature to prepare tazobactam starting from 6-APA. According to the patent by Shiyong reported in Figure 4 [145], tazobactam was obtained performing first the deamination of 6-APA under the action of H_3_PO_2_ and NaNO_2_/HCl to obtain **1**, followed by the oxidation of the sulfur atom of **1** into sulfoxide group using ceric triflate/hydrogen peroxide [Ce(OTf)_4_/H_2_O_2_] as an oxidant system to obtain compound **2**.

Compound **2** was treated with trifluoroacetic anhydride (TFAA) and was refluxed at 95–105 °C under reduced pressure to obtain the acylated compound **3**, which reacted with hydrazine hydrate to give the reduced compound **4**. Next, using triethylamine (Et_3_N) as an acid binding agent, **4** was treated with methane sulfonyl chloride to obtain the sulfonated compound **5**. By condensing **5** and triazole in the presence of Et_3_N to achieve **6** and oxidating **6** with hydrogen peroxide and acetic anhydride (Ac_2_O) in the presence of sodium polyphosphate to obtain a sulfone group, tazobactam was obtained as free acid.

According to the authors, the advantages of this procedure consist of simple operation steps and of low-cost raw materials being easy to obtain. Additionally, the reaction steps are short, and the obtained intermediate products have high purity. The final product of tazobactam acid obtained as white solid powder showed a purity of more than 99.5% and the high total molar yield makes this procedure suitable for industrial production. Very recently, Shuhao et al. have reported a combination of continuous flow and batch experiments for the synthesis of tazobactam, which was prepared as described in Figure 5 [146]. The continuous flow method has not only greatly reduced the reaction time, but also significantly improved procedure safety and increased the yield. 

Briefly, 6-APA and potassium bromide were added to an aqueous solution of sulfuric acid and dichloromethane (DCM) at 0 °C, and the mixture was added with an aqueous solution of sodium nitrite. After the reaction, the organic phase containing the 6-bromo derivative of 6-APA (6-BrPA), was washed with saturated brine, treated with 1-(3-dimethylaminopropyl)-3-ethylcarbodiimide hydrochloride (EDC), diphenylmethanol, and dimethyl amino pyridine (DMAP) to realize the protection of the carboxylic group, and the product was directly treated without separation with 50% hydrogen peroxide, acetic acid, sulfuric acid, and ethylenediaminetetraacetic acid (EDTA). The 20% peroxyacetic acid formed oxidized the sulfur atom of the protected 6-BrPA to sulfoxide group. The subsequent treatment with zinc and ammonium acetate (NH_4_Ac) provided the 6,6-dihydropenicillin-3-α-carboxylic acid as white solid, which was transformed in the 3-methyl-[2-oxo-4-(2-benzothiazole di-thio)-1-azacyclobutyl]-3-butenoic acid by treatment with 2-mercaptobenzothiazole in toluene, refluxing for 80 min. The obtained compound was then reacted with CuBr_2_ to obtain the 2-β-bromomethyl-2-α-methylpenicillin-3-α-carboxylic acid diphenyl methyl ester which was transformed in β-(1,2,3-triazol-1-yl) methyl-2α-methyl-6,6-dihydropenicillin-3-α-carboxylic acid diphenyl methyl ester by reacting it with 1,2,3-triazole in acetone/water. The tazobactam precursor, namely 2-β-(1,2,3-triazol-1-yl) methyl-2-α-methyl-6,6-dihydropenicillin-3-α-carboxylic acid diphenyl methyl 1,1-dioxide, was obtained in DCM and acetic acid (AcOH) under treatment with potassium permanganate. Tazobactam was finally obtained, deprotecting the previous compound dissolved in m-cresol with 15% hydrochloric acid for 3 h, and then filtered.

##### Synthesis of Tebipenem Pivoxil

Initially researched and developed by Pfizer Inc. (New York City, NY, USA), tebipenem pivoxil developed by Japanese Mingzhi company obtained Japan’s approval and went on Japan’s market in April 2009. Tebipenem pivoxil is the orally available prodrug of tebipenem, which is hydrolyzed in vivo by esterase enzymes and, even if only in Japan, it is the only orally administrable carbapenem marketed. While acting as BLAs by binding PBPs, and inhibiting the synthesis of bacteria cell wall, tebipenem also works as inhibitor of BLEs produced by *M. tuberculosis*. The improved synthesis of tebipenem pivoxil was described by Peng and co-workers and has been herein shown in Figure 6 [147]. 

Briefly, tebipenem pivoxil was synthesized from commercially available materials as (4-nitrophenyl)methyl (4R,5R,6S)-3-[(diphenoxyphosphinyl)oxy]-6-[(1R)-1-hydroxyethyl]-4-methyl-7-oxo-1-azabicyclo [3.2.0] hept-2-ene-2-carboxylate **1** and 3-mercapto-1-(1,3-thiazolin-2-yl) azetidine **2** via a substitution reaction in presence of ethyl-di-isopropyl-amine [EtN(iPr)_2_] in acetonitrile (MeCN), which afforded compound **3**. Its subsequent hydrogenation under H_2_ pressure in butanol (BuOH) to remove the p-nitro-benzyl protecting group followed by esterification with 2,2-dimethyl-propionic acid iodo-methyl ester using benzyl triethylammonium chloride (TEBAC) afforded tebipenem pivoxil with a total yield of 56.7%. This process was prone to operating in high yield and it provided a basis for a pilot scale production.

#### 5.1.2. Not Yet Clinically Approved β-Lactam BLEsIs

##### Synthesis of Enmetazobactam

Since it is structurally like tazobactam, except for the presence of an additional methyl group in the triazole ring, enmetazobactam can be prepared directly upon methylation of tazobactam. Three procedures have been reported in literature to perform the methylation of the triazole ring [148,149,150]. According to the procedure described in the patent by Faini et al. [150], enmetazobactam was prepared from tazobactam via silylation/esterification, with N,O-bis(trimethylsilyl)acetamide in DCM followed by quaternization via N-alkylation of the obtained intermediate with methyl triflate and subsequent desilylation with sodium 2-ethylhexanoate in EtOH (Figure 7).

Similar procedures that have been slightly modified have been reported by Senthilkumar et al. and by Palanisamy et al. [148,149]. Notably, in a first step, a tazobactam suspension in acetone at 25–30 °C was added slowly with N,O-bis(silyl)acetamide and then with methyl iodide in place of methyl triflate to obtain the quaternization of the triazole ring. Then, the afforded solid product was, in a second step, treated with sodium thiosulfate in water, and after the due purification procedures using a solution of Amberlite LA-2 resin, DCM washings, and activated carbon, enmetazobactam was obtained upon lyophilization.

##### General Procedure to Synthetize 6-Methylidene-penem Carboxylic Acid Sodium Salts

Various 6-methylidene-penem carboxylic acid sodium salts were achieved according to Figure 8 [151,152].

Briefly, 6-APA was converted to 6(S)-bromo penicillanic acid **1** by treatment with NaNO_2_ in the presence of HBr. Then, the esterification of the di-cyclohexylamine salt of **1** with p-methoxybenzyl bromide, followed by oxidation with m-chloro-perbenzoic acid (MCPBA), afforded the sulfoxide **2**. Sulfoxide **2** was then heated to reflux in toluene with 2-mercaptobenzothiazole to firstly achieve the sulfenic acid which, reacting with the mercaptan, provided the disulfide **3**. Compound **3** was then transformed by a base-catalyzed double bond isomerization to the conjugated ester disulfide **4**, which was in turn transformed in the formyl-thio-derivative **5** by reductive formylation. By ozonolysis of **5,** the oxalamide **6** was obtained, which underwent cyclisation to provide the crystalline p-methoxybenzyl (SR,6S)-6-bromopenem-3-carboxylate **7**, by heating a toluene solution of **6** at 95 °C for 30 min in the presence of excess trimethyl phosphite [P(MeO_3_)]. Subsequently, involving a Lewis acid (anhydrous MgBr_2_), an aldol condensation reaction was performed between several selected aldehydes and **7** in the presence of Et_3_N as the base. The resulting bromohydrins **8** were trapped in the form of their respective acetylated derivatives **9**.

Then, for performing the reductive elimination procedure to introduce a Z-double bond between the C6 and C7 on the obtained acetoxy bromohydrins, a neutral procedure was devised using freshly prepared activated zinc and 0.5 M (pH 6.5) phosphate buffer (PBS) at room temperature in acetonitrile–THF (1:2). The resultant crude products were purified by Diaion HP-21 resin (80 mL, Mitsubishi Kasei Co. Ltd.) column chromatography. Finally, following Lewis’s acid-mediated deprotections of the ester compounds and treatments with Na_2_HPO_4_ aqueous solutions, the correspondent sodium salts were obtained.

##### Synthesis of LN-1-255

Among other penicillin sulfones, which structurally incorporate both a 6-position alkylidene substituent and a 2′-β-substituent, LN-1-255 has been synthesized by Buynak et al. according to Figure 9 [153].

Firstly, 6-APA was treated with EDC, diphenylmethanol, and DMAP to protect the carboxylic group and obtain compound **1**, which was transformed in α-diazo-β-lactam by means of 2-nitropropane in trifluoro acetic acid (TFA) and then into the α-oxo-β-lactam **2** using catalytic amounts of rhodium (II) in the presence of the extremely mild oxygen donor, propylene oxide. The obtained α-oxo-β-lactam **2** underwent a Wittig reaction using the triphenyl-pyridin-2-yl-methyl-phosphonium chloride and NaNH_2_. The oxidation of the 6-methylidene pyridine derivative **3** with MCPBA afforded the corresponding sulfoxide, which was heated to reflux in toluene with 2-mercaptobenzothiazole, obtaining the corresponding disulfide. The disulfide was reacted with 3,4-di-tert-butoxy-phenyl-acetic acid and silver acetate, and after removal of the t-butyl protecting groups by TFA treatment and the benzyl protecting group by catalytic hydrogenation, LN-1-255 was achieved. By treatment with Na_2_HPO_4_ aqueous solutions, the corresponding sodium salt was achieved.

### 5.2. Non-β-Lactam DBO BLEsIs

Non-β-lactam BLEsIs named DBOs consist of molecules which do not possess a β-lactam ring but have a five-membered DBO ring and an amide group, which is the responsibility of the interaction with the active-site of SBLEs by a carbamylation reaction. While the β-lactam BLEsIs interact irreversibly with the enzymes, DBOs do not. Particularly, DBOs bind reversibly to the enzyme in the active site and undergo the opening of the DBO ring forming carbamate derivatives, following a de-acylation process to regenerate the original compounds, which can inactivate new enzymes.

#### 5.2.1. Clinically Approved DBO BLEsIs

##### Synthesis of Avibactam

The first patents describing the synthesis of avibactam were published in 2002 (international version) and in 2003 (US version). The original syntheses of avibactam do not provide details on the absolute configuration of the molecule. Subsequently, the synthesis of avibactam starting from chiral derivatives was described. Chiral piperidines, L-glutamic acid, and L-pyroglutamic acid were considered as starting materials and Boc-benzyl-L-glutamate was used to optimize the synthetic process [154]. Recently, Laure Peilleron and Kevin Cariou have reported the synthesis of avibactam starting from a commercially available chiral oxo-pyrrolidine (Figure 10) [155] (Novexel).

Briefly, Novexel chemists started the synthesis of avibactam from commercially available enantiopure 2-pyrrolidone (γ-lactam), having Boc and benzyl protecting groups. First, the γ-lactam was subjected to Corey–Chaykovsky-type reaction conditions in a mixture dimethyl sulfoxide/tetrahydrofuran (DMSO/THF) as solvent, and the addition of a sulfonium ylide yielded the zwitterionic enolate **1**. The reaction of **1** with lithium hydroxide (LiOH) and methane sulfonic acid (MsOH) gave an intermediate chloro-ketone onto which O-benzylhydroxylamine (BnONH_2_) was condensed to furnish the E/Z oxime **2**. The cyclization to piperidine **3** was followed by acidic treatment to remove the Boc protecting group and by the reduction of oxime to hydroxylamine derivative that was isolated as its oxalate salt **4** (75/25 trans/cis isomers). The bicyclic core was generated by treatment with tri-phosgene to form the ureic derivative **5**. A rigorously controlled ester saponification with LiOH allowed to hydrolyze selectively the trans derivative, thus solely obtaining the corresponding trans carboxylic acid, which was converted to the primary amide **6** by treatment with pivaloyl chloride followed by ammonia. Then, after debenzylation by hydrogenolysis, the sulfate group was introduced, and avibactam was isolated first as tetrabutylammonium salt and then as sodium salt. Subsequently, the transformation of the starting material into piperidine was optimized using potassium tert-butoxide (t-BuO^−^K^+^) in place of sodium hydride, performing the chlorination and the formation of the oxime in a one-pot fashion, and purifying only the final piperidine. Since only the trans isomer of piperidine **3** was isolated, only the R isomer of **4** was obtained after reduction, therefore the selective saponification step could be avoided, and the amide derivative was directly obtained from **4**. Subsequently, a selective fluorenyl-methyl-oxy-carbonyl (Fmoc) protection of the amine was necessary before activating the hydroxylamine with carbonyl di-imidazole to give an intermediate which, upon deprotection of the Fmoc group, underwent cyclization in the presence of di-ethylamine to give the bicyclic adduct **7**.

##### Synthesis of Relebactam

The scale-up synthesis of relebactam has been described in Figure 11 [156].

The synthesis comprised the sulfonylation of 5-hydroxypiperidine-2-carboxylic acid with 2-nitrobenzene sulfonyl chloride (o-NsCl) followed by an intramolecular esterification which provided bridged ring compound **2**. Next, a ring-opening reaction involving the Boc-protected amino-piperidine followed by treatment with o-NsCl and DMAP generated the corresponding arene-sulphonate **3**. Compound **3** was then treated with N-benzyl-oxybenzsulfamide to give the corresponding product of substitution, namely **4**. Treatment of **4** with thioglycolic acid gave **5**, which was the key intermediate of this synthesis. By the treatment of **5** with tri-phosgene in the presence of N,N-diisopropylethylamine (DIPEA), cyclization occurred, generating a chiral bridged ring intermediate **6.** Then, **6** was treated with catalytic Pd/Al_2_O_3_ under hydrogen gas in THF to remove the benzyl protecting group and obtain **7**, which was in turn sulfated using tetrabutylammonium bis-sulphate as a sulfate radical source and 2-picoline as a base in THF. After sulfate formation and N,O-bis(trimethylsilyl)trifluoroacetamide (BSTFA) and trimethylsilyl iodide (TMSI) mediated N-Boc deprotection, treatment with acetic acid promoted acidic precipitation of relebactam.

#### 5.2.2. Not Yet Clinically Approved DBO BLEsIs

##### Synthesis of Zidebactam

In addition to various patents, the synthesis of zidebactam has been described by Papp-Wallace and co-workers as reported in Figure 12 [101].

Briefly, starting from trans-6-benzyloxy-7-oxo-l,6-diaza-bicyclo [3.2.1] octane-2-carboxylic acid sodium salt, which is an intermediate of the synthesis of avibactam previously described, compound **1** was obtained using N,N-dimethyl formamide (DMF), EDC hydrochloride, butyl alcohol, and (R)-N-tert-butoxycarbonyl-piperidin-3-carboxylic acid hydrazide. Then, once **1** was obtained as a white solid, it was dissolved in methanol and treated with 10% palladium on carbon under pressure of hydrogen to provide **2** as a pale pink solid, which was used for the next reaction immediately. Therefore, **2** was treated with pyridine sulfur trioxide complex, in pyridine in the presence of a 0.5 N aqueous potassium dihydrogen phosphate solution and tetrabutylammonium sulphate to provide **3** as a yellowish solid, in quantitative yield. Finally, zidebactam was obtained as free sulfonic acid by treating **3** with TFA in DCM to remove the Boc protecting group.

##### Synthesis of WCK 5153

WCK5153 (C_12_H_19_N_5_O_7_S, MW = 377.1), IUPAC name, (2S,5R)-7-oxo-2-(2-((S)-pyrrolidine-3-carbonyl) hydrazine-1-carbonyl)-1,6-diazabicyclo [3.2.1] octan-6-yl hydrogen sulfate, was synthetized as zidebactam using (R)-N-tert-butoxycarbonyl-pyrrolidin-3-carboxylic acid hydrazide in place of (R)-N-tert-butoxycarbonyl-piperidin-3-carboxylic acid hydrazide according to Papp-Wallace et al. [101].

##### Synthesis of WCK-4234

Figure 13 represents the synthesis of WCK-4234 [101].

The trans-6-benzyloxy-7-oxo-l,6-diaza-bicyclo [3.2.1] octane-2-carboxylic acid sodium salt was converted into the anhydride **1** by treating the sodium salt with triethylamine hydrochloride (TEA) in DCM and reacting the resulting compound with pivaloyl chloride in the presence of TEA at 0−5 °C. Compound **1** was reacted as such with a 25% aqueous solution of ammonia in water at −20 °C to obtain the amide **2** as an off-white solid after workup and purification. The amide was dehydrated with trifluoroacetic anhydride (TFAA) in the presence of TEA in DCM to obtain the cyano derivative **3**, which was debenzylated with 10% Pd/C in a 1/1 mixture of DMF/DCM under hydrogen atmosphere to obtain the hydroxyl compound **4**. This compound was then immediately sulfated with the DMF:SO_3_ complex to obtain the sulfate, which was isolated as its tetrabutylammonium salt **5** by reacting with tetrabutylammonium acetate. The tetrabutylammonium salt was converted to WCK-4234 in the form of sodium salt by passing **5** through a column filled with Indion 225 sodium resin.

##### Synthesis of ETX-1317 and of Its Orally Administrable Prodrug ETX-0282

The gram-scale synthesis of ETX-0282 and ETX-1317 was reported by Durand-Réville and colleagues and has been presented in Figure 14 [97].

The chiral isopropyl and ethyl bromo-fluoroacetates **I** and **II** were first prepared in three steps from racemic ethyl bromo-fluoroacetate by saponification with NaOH, followed by chiral salt resolution using (S)-1-phenylethanamine. The chiral salt was then re-esterified using isopropanol or ethanol to give the (R)-isopropyl and the desired ethyl bromo-fluoro-acetate. The key intermediate of the synthesis of ETX0282 and ETX1317 was the hydroxyurea **III**, which was prepared in 10 steps from commercially available materials. Briefly, ethyl 2-oxoacetate and (S)-2-methylpropane-2-sulfinamide were condensed to afford the chiral imine **1**. An aza-Diels−Alder reaction with isoprene provided compound **2**, which underwent deprotection to remove the tert-butyl sulfinyl group to afford **3**, subsequently Boc protected, to give compound **4**. The saponification of the ester followed by amide coupling using 1,1′-carbonyldiimidazole (CDI) and ammonium acetate afforded compound **5**. Among the many strategies developed to install the hydroxylamine functionality in a regio- and stereo-selective manner, the reaction of alkene **5** with N-Boc-hydroxylamine in the presence of oxygen or air gave the desired compound **6** in a single step and in 40% yield. Compound **6** was then protected with a tert-butyl-silyl (TBS) group using TBS chloride (TBSCl) to afford **7**, which was Boc deprotected using zinc bromide obtaining compound **8**. Cyclization of the diamine **8** with tri-phosgene provided the corresponding cyclic urea **9**, which was TBS deprotected with HF-pyridine (HFPy) to give the key intermediate **III**. By alkylation of **III** with **I** or **II**, in the presence of 1,8-diazabiciclo [5.4.0] undec-7-ene, (DBU), ETX-0282 and compound **10** were obtained, respectively, while ETX1317 was prepared in 63% yield from **10** by saponification with lithium hydroxide.

##### Synthesis of Durlobactam

Chemically, durlobactam is [(2S,5R)-2-carbamoyl-3-methyl-7-oxo-1,6-diazabicyclo [3.2.1] oct-3-en-6-yl] hydrogen sulfate which can be prepared from the key intermediate hydroxyurea 6-hydroxy-3-methyl-7-oxo-1,6-diaza-bicyclo [3.2.1] oct-3-ene-2-carboxylic acid amide **I**, which is the structural isomer of **III** prepared to synthetize ETX-1317 [101]. Then, according to Figure 15, compound **1** obtained in the synthesis of **III** (Figure 14) was reacted with penta-1,3-diene in place of isoprene, and, by an aza-Diels−Alder reaction, compound **2** was obtained.

Compound **2** underwent deprotection of the tert-butyl sulfinyl group to afford **3**, subsequently Boc protected, to give compound **4**. The saponification of the ester followed by amide coupling using ammonium acetate afforded compound **5**. The reaction of alkene **5** with N-Boc-hydroxylamine in the presence of oxygen or air gave the desired compound **6** in a single step. Compound **6** was then protected with TBS group, using TBSCl to afford **7**, which was Boc deprotected using zinc bromide obtaining compound **8**. Cyclization of the diamine **8** with tri-phosgene provided the corresponding cyclic urea **9**, which was TBS deprotected with HFPy to give the key intermediate **I**. This compound was then immediately sulfated with the DMF:SO_3_ complex to obtain the sulfate, which was isolated as its tetrabutylammonium salt **10** by reacting with tetrabutylammonium acetate. The tetrabutylammonium salt was converted to durlobactam in the form of sodium salt by passing **10** through a column filled with Indion 225 sodium resin.

##### Synthesis of ANT-3310

The syntheses of various DBOs have been recently reported by Davies and colleagues, starting from the commercially available 5-benzyloxyamino-piperidine-2-carboxylic acid ethyl ester—oxalate salt [127]. With the same procedure, it is also possible to start using the compound **4** obtained during the synthesis of avibactam, as reported in the Figure 16.

Briefly, the benzyl protected diamine oxalate salt **4**, prepared as shown in Figure 10, was converted to the free base **1** using potassium hydrogen carbonate, which was cyclized to the bicyclic urea **2** with tri-phosgene. The subsequent saponification of compound **2** using LiOH provided the key intermediate **3**. The following decarboxylation and radical treatment of **3** with Selectfluor in the presence of AgNO_3_ gave the fluoro derivative **4a**, which was transformed in compound **5** by hydrogenolysis to remove benzyl protecting group. Compound **5** was then submitted to the usual chemistry sequence described previously to achieve other DBOs for obtaining ANT-3310.

##### Synthesis of Nacubactam

Performing a procedure described in a patent by Maiti and co-workers, nacubactam can be prepared according to Figure 17 [157].

Briefly, starting from the key intermediate **3**, obtained during the previous synthesis, and using the commercially available 2-(amino-oxy-ethyl) carbamic acid tert-butyl ester **1** in the presence of benzotriazolol and EDC hydrochloride as condensing agents, 4-DMAP as catalyst, and DCM as solvent, the compound **2** was obtained. Then, compound **2** was submitted to the usual chemistry sequence described previously, including debenzylation and treatment with pyridineSO3 in pyridine, to achieve **4**. The subsequent removal of Boc protecting group with TFA provided nacubactam in the form of trifluoroacetate salt.

##### Synthesis of GT-055

According to our knowledge, up to today, no synthetic procedure has been reported in literature to prepare GT-055 (also referred to as LCB18-055), responding to the IUPAC name: sulfuric acid mono-[2-(5,5-bis-aminomethyl-4,5-dihydro-isoxazol-3-yl)-7-oxo-1,6-diaza-bicyclo [3.2.1] oct-6-yl] ester; trifluoro-acetate salt (C₁₃H₂₀F₃N₅O₈S, MW = 463.4).

### 5.3. Non-β-Lactam Boronic Acid Derivatives

Non-β-lactam boronic acid derivatives, developed to afford non-acylating BLEsIs, are cyclic derivatives of boronic acid, which can react very rapidly with BLEs, forming stable enzyme-inhibitor complexes.

#### 5.3.1. Clinically Approved Boronic Acid Derivative

##### Synthesis of Vaborbactam

The synthesis of various cyclic boronates, including vaborbactam, has been reported by Scott and co-workers, as shown in Figure 18 [110].

Briefly, vaborbactam was prepared in a six-step procedure in an overall yield of about 30% starting from the enantiopure β-hydroxy ester **1**, in turn prepared by lipase-mediated kinetic resolution of the corresponding racemate. Particularly, compound **1** was protected as silyl ether with tert-butyl-dimethylsilyl chloride (TBDSCl) and imidazole, achieving compound **2**. Its regioselective hydroboration catalyzed by [Ir(COD)Cl]_2_ gave the pinacol boronate **3**, which was converted to the more stable pinanediol boronate **4**. Its stereoselective chloromethylation following Matteson’s protocol at −95 °C, using butyl lithium (n-BuLi) and DCM afforded the (S)-chlore homologue **5** as an 85:15 mixture of diastereomers. Then, the displacement of the chlorine group with lithium hexamethyl disilazide (LiHMDS) followed by in situ acylation with the thiophen-2-yl-acetic acid gave the acyl amido boronate **6**. The acidic removal of all protecting groups afforded vaborbactam.

#### 5.3.2. Not Yet Clinically Approved Boronic Acid Derivatives

##### Synthesis of Taniborbactam

According to Liu et al., taniborbactam was prepared as shown in Figure 19 [158].

Starting from the commercially available 3-borono-2-methoxybenzoic acid **1** and using isobutylene, followed by the esterification of the boronic acid with (+)-pinanediol, compound **2** was prepared. Following Matteson’s protocol, the aryl boronate **2** was reacted with chloromethyl lithium at −100 °C to provide **3** in excellent yield. A second homologation reaction with the anion derived from DCM (−100 °C) selectively provided the (S)-α-chloro-boronate **4**. The stereospecific displacement of the chlorine atom of **4** with lithium bis(trimethylsilyl)amide at −20 °C afforded the intermediate α-silyl amino boronate **5**, which was coupled in situ using 1-[bis (dimethyl amino)methylene]-1H-1,2,3-triazolo [4,5-b] pyridinium 3-oxide hexafluorophosphate (HATU) and N-methyl morpholine (NMM) to the requisite carboxylic acid **I**, in turn prepared as reported in Figure 20. The resultant amide **6** was deprotected and cyclized by treatment with BCl_3_ (1 M in DCM) at −78 °C to afford the crude boronate **7**, which was purified by reverse-phase high-performance liquid chromatography (HPLC) and then lyophilized to dryness.

The carboxylic acid **I** was prepared starting from the commercially available (4-tert-butoxycarbonylamino-cyclohexyl)-acetic acid **1**, by its alkylation with benzyl bromide to give the ester derivative **2**. The removal of the Boc protective group afforded the amine **3** as HCl salt, which was converted to the intermediate **4** via reductive amination. The protection of the secondary amine and the removal of the benzyl group provided **I** in a 30% yield over five steps.

##### Synthesis of VNRX-7145

VNRX-7145, also known as ledaborbactam etzadroxil, which is the orally bioavailable prodrug of ledarbobactam (VNRX-5236), was prepared as shown in Figure 21 using the intermediate **4** of the previous synthesis as a starting material [120].

Particularly, compound **4**, isolated from two consecutive homologation reactions following Matteson’s protocol, was treated with lithium bis(trimethylsilyl)amide (LHMDS) at −20 °C to afford the desired stereoisomer intermediate **1**. The α-silylaminoboronate intermediate **1** was treated in situ with HATU and NMM and coupled with the propyonic acid to achieve compound **2**. Then, **2** was treated with TFA in DCM or 4 N HCl in dioxane to provide acid **3**. Reaction of **3** with sodium carbonate, sodium iodide, and chloromethyl etzadroxilate in DMF afforded ester **4a**. Finally, the pinanediol and methyl ether were selectively removed by treatment with aluminium chloride in DCM to provide the desired compound VNRX-7145.

##### Synthesis of Xeruborbactam

The synthesis of xeruborbactam has been recently reported by Hecker and colleagues, as shown in Figure 22 [159].

Briefly, the commercially available 2-bromo-5-fluorophenol **1** was converted to the Boc-protected derivative **2** using di-tert-butyl decarbonate (Boc_2_O) and DMAP.

Subsequently, compound **2** was treated with lithium di-isopropyl-amide (LDA) to achieve deprotonation and then with di-tert-butyl decarbonate to obtain the intermediate **3**. The two protecting groups were replaced with a single acetonide in compound **4**, which was obtained treating **3** with TFA and acetone. Compound **4** was subjected to a Heck reaction using palladium acetate and chloro-[tri(o-tolyl) phosphine] [2-(2′-amino-1,1′-biphenyl)] palladium (II) (P(o-toly)_3_) to introduce the acrylate side chain, giving **5**. Bromination of **5** followed by decarboxylative elimination afforded cis-vinyl bromide **6**, which underwent palladium-mediated borylation with the bis[(+)-pinanediolato] di-boron affording the cis-vinyl boronate **7**. Compound **7** was subjected to palladium-catalyzed cyclopropanation with diazomethane to afford **8** as a mixture of two diastereomers, which were separated by HPLC. The subsequential hydrolysis of the acetonide with NaOH and of the pinanediol protecting group with TFA in the presence of triethyl silane (TES) and B-tert-butyl boronic acid [t-BuB(OH_2_)] afforded xeruborbactam.

### 5.4. Non-β-Lactam Thiazolyl Acid Derivatives

Non-β-lactam thiazolyl acid derivatives are small molecules that are structurally different from the other novel inhibitors previously described. Among them, ANT-2681 contains a thiazole carboxylate derivative consisting of a 3,5-difluoro benzensulfonamide nucleus linked to a 4-thiazol-carbammide in the position 5 of the heterocycle with an ureidoguanidine residue in position 4.

#### Synthesis of the Not Yet Approved ANT-2681

ANT-2681 was prepared by Davies and his group according to Figure 23 [160].

The commercially available methoxybenzyl isothiocyanate **1** was treated with a solution of tert-butyl iso-cyanoacetate previously added to a suspension of potassium tert-butoxide in dry tetrahydrofuran, achieving compound **2** as a pale-yellow solid. Subsequently, **2** was added to a suspension of NaH and then treated with a solution of 4-bromo-3,5-difluoro-benzenesulfonyl chloride in THF achieving **3** as a pale-yellow solid again. By treating a mixture of **3**, Xantphos, tris (dibenzylidene acetone) dipalladium (0) [Pd_2_(dba)_3_] and K_3_PO_4_ with a saturated solution of NH_3_ in dioxane was afforded **4**, which was transformed in the unstable nitrophenyl carbamate **5** by treatment with (4-nitrophenyl) chloroformate in toluene at room temperature. The nitrophenyl carbamate was used in the next step without further purification. Therefore, a solution of compound **5** in THF was reacted with DIPEA affording **6** as a pale-yellow solid, which was converted in the intermediate **7** using HCl in Et_2_O. **7** was obtained as an off-white solid, which was immediately used without purification. The treatment of **7** with DIPEA and N,N’-bis(tert-butoxy carbonyl)-1H-pyrazole-1-carboxamidine in DMF afforded **8**, which was converted to the desired product ANT-2681 using TFA at room temperature. With this step, ANT-2681 was isolated as zwitterion, which was purified by preparative HPLC. Subsequently, ANT-2681 was isolated as sodium salt by treatment with 0.05N aqueous NaOH at room temperature in deionized water.

## 6. Conclusions and Authors Considerations

By now, it is established that the defensive strategy of bacteria against β-lactam antibiotics (BLAs) mainly consists of producing β-lactamase enzymes (BLEs), while the current approach of scientists to rehabilitate antibiotics that are no longer functioning and counteracting infections that are increasingly difficult to treat consists of developing β-lactamases inhibitors (BLEsIs). In the context of this work, after having briefly provided the basic information about the most clinically relevant BLAs and BLEs, we have originally examined BLEsIs by a chemical approach. Particularly, we have herein reviewed both the already clinically approved BLEsIs, and those yet under early stages, preclinical, and clinical investigations developed so far, reporting the long-established synthetic procedures, as well as those recently optimized and still to be optimized present in literature.

Unfortunately, along with our investigation to produce this work, it has been evidenced that, while both the number of BLEs-producing bacteria and that of BLEs themselves are rapidly and constantly evolving, the number of BLEsIs clinically approved and available for therapeutic uses is evolving very slowly. The genes encoding for bacterial BLEs are easily transmitted between bacteria, especially if linked to plasmids, thus continuously increasing the number of pathogens producing BLEs. Additionally, these enzymes change continuously, giving rise to an increasing number of new variants of BLEs with an expanding broader spectrum of action and the capability of inactivating even the last generation associations BLAs/BLIs.

The alarming scenario is that of an unequal struggle between a small number of inhibitors and an immense and increasing number of different types of BLEs, progressively potent, which make the researchers’ effort to create substances capable of covering their entire spectrum of action almost utopic. Nowadays, among the BLEsIs researched, developed, and in development, the clinically approved ones are only 27%, while most of them are still in the preclinical or clinical trials (63%), or even at very early stages of development (10%).

In Europe and the USA, only six are the products currently on the market, and among these, no compound is active on class B MBLEs, including NDM-1, which is able to hydrolyze all available antibiotics except for aztreonam. The long-established β-lactam inhibitors are inactive on carbapenemases, which can inactivate the most resistant carbapenems of last generation. Only three compounds recently developed, two belonging to the DBOs compounds and one making part of the boronic acid derivatives, inhibit class A carbapenemases (KPC), while only the DBO avibactam is active on class D OXA-like carbapenemases.

Promisingly, higher percentages of associations BLAs/BLEsIs are active on class D OXA-like carbapenemases (56.4%) and on class A carbapenemases of KPC family (74.5%), if not yet approved BLEsIs are also considered. Particularly, although the percentages of compounds active against most class D carbapenemases and above all against carbapenemases of class B is still low, almost all the new not clinically applied BLEsIs in development are capable to inhibit KPCs, and more than 50% work well against OXA-48.

This situation highlights that an increasing effort to synthetize a progressively higher number of differently structured molecules also capable of inhibiting the most recent and potent variants of BLEs produced by the most clinically relevant bacteria is necessary. To this end, a successful approach consists of considering molecules or chemical nuclei already known for being active as BLEsIs as templates and making structural changes to improve their potency and spectrum of activity. Therefore, we think that having provided with this review detailed information concerning the most accredited synthetic procedures to prepare the key molecules of each structural category to date investigated as BLEsIs, this could be of great use for pharmaceutical chemists operating in the sector.

## Data Availability

Data is contained within the article.

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
