# Peer review of "Recommendations to Synthetize Old and New β-Lactamases Inhibitors: A Review to Encourage Further Production"

_pharmaceuticals, 2022, doi:10.3390/ph15030384_

Round 1
Reviewer 1 Report
Very interesting and comprehensive review.
The authors have chosen an unusual approach by proposing the synthesis pathways of beta-lactamases.
There are therefore several possible reading levels depending on the knowledge of the reader. The first part will be of interest to all audiences; however the “Guidelines” part will be accessible to more specialized people.
Concerning the title of the article: Guidelines is a strong term, the term recommendation seems more appropriate to me.
Author Response
Very interesting and comprehensive review.
The authors have chosen an unusual approach by proposing the synthesis pathways of beta-lactamases.
There are therefore several possible reading levels depending on the knowledge of the reader. The first part will be of interest to all audiences; however the “Guidelines” part will be accessible to more specialized people.
Concerning the title of the article: Guidelines is a strong term, the term recommendation seems more appropriate to me.
We are very grateful to the Reviewer for his kind comments and appreciation of our approach to the topic of the present review.
We also agree with his comment on the title which has been changed accordingly.
Reviewer 2 Report
This manuscript systematically summarizes the research progress of β-lactamases inhibitors. Based on the fully understanding of pharmaceutical chemistry of β-lactam antibiotics and β-lactamase enzymes, the authors reviewed various studies of β-lactamases inhibitors, especially summarizing them through several intuitive and concise charts. It can be said that this is not so much a review as a textbook chapter on β-lactamases inhibitors. The manuscript has substantial contents and should be of great interest to medical personnel.
Author Response
This manuscript systematically summarizes the research progress of β-lactamases inhibitors. Based on the fully understanding of pharmaceutical chemistry of β-lactam antibiotics and β-lactamase enzymes, the authors reviewed various studies of β-lactamases inhibitors, especially summarizing them through several intuitive and concise charts. It can be said that this is not so much a review as a textbook chapter on β-lactamases inhibitors. The manuscript has substantial contents and should be of great interest to medical personnel.
Based on the Reviewer comments, it seems to us that we do not have to do anything more than what we have done so far to improve our work. It is not so?
Reviewer 3 Report
This manuscript is an interesting, well written review on β-lactam antibiotics (BLAs) and β-lactamase enzymes inhibitors (BLEsIs) with emphasis on the synthetic procedures for the preparation of BLEsIs.
It can be accepted after the following minor corrections:
Line 71: please replace growth by grow
Line 72: replace include by includes
Line 172: replace achieve by afford
Line 417: an additional scheme describing this procedure for the synthesis of tazobactam could be included
Line 513: replace achieved by obtained
Line 523: replace “to realize the protection” by “to protect”
Lines 530, 531 and 534: replace correspondent by corresponding
Scheme 9: replace pyvaloyl by pivaloyl
Line 574: replace get by give
Line 580: replace sulfoonylation by sulfonylation
Line 612: rephrase “Finally, zidebactam was obtained as free sulfonic acid removing the Boc protecting group by treating 3 in DCM with TFA” as follows
“Finally, zidebactam was obtained as free sulfonic acid, by treating 3 with TFA in DCM to remove the Boc protecting group”
Lines 615 and 616 “(C12H19N5O7S, MW = 377.1), IUPAC name, (2S,5R)-7-oxo-2-(2-((S)-pyrrolidine-3-carbonyl)hydrazine-1-carbonyl)-1,6-diazabicyclo [3.2.1] octan-6-yl hydrogen sulfate,” :
this could appear only in Scheme 12 legend
Line 650: replace “and ethyl bromo-fluoro-acetate desired.” By “and the desired ethyl bromo-fluoro-acetate.”
Lines 675, 695 and 805: replace achieved by obtained
Line 747: replace achieved by prepared
Line 818: replace “and its group” by “and his group”
Line 827: “was removed” delete “was”
Line 846: replace “In this contest, in this work,” by “In the context of this work”
Author Response
This manuscript is an interesting, well written review on β-lactam antibiotics (BLAs) and β-lactamase enzymes inhibitors (BLEsIs) with emphasis on the synthetic procedures for the preparation of BLEsIs.
It can be accepted after the following minor corrections:
Line 71: please replace growth by grow
Done
Line 72: replace include by includes
Done
Line 172: replace achieve by afford
Done
Line 417: an additional scheme describing this procedure for the synthesis of tazobactam could be included
The required scheme has been included as Scheme 4b.
Line 513: replace achieved by obtained
Done
Line 523: replace “to realize the protection” by “to protect
Done
Lines 530, 531 and 534: replace correspondent by corresponding
Done
Scheme 9: replace pyvaloyl by pivaloyl
Done
Line 574: replace get by give
Done
Line 580: replace sulfoonylation by sulfonylation
Done
Line 612: rephrase “Finally, zidebactam was obtained as free sulfonic acid removing the Boc protecting group by treating 3 in DCM with TFA” as follows
“Finally, zidebactam was obtained as free sulfonic acid, by treating 3 with TFA in DCM to remove the Boc protecting group”
Done
Lines 615 and 616 “(C12H19N5O7S, MW = 377.1), IUPAC name, (2S,5R)-7-oxo-2-(2-((S)-pyrrolidine-3-carbonyl)hydrazine-1-carbonyl)-1,6-diazabicyclo [3.2.1] octan-6-yl hydrogen sulfate,” :
this could appear only in Scheme 12 legend
We make kindly note to the Reviewer that the information contained in lines 615-616 is different from that contained in the caption of Scheme 12. In fact, it refers to compound WCK-5153, whose scheme has not been reported because practically identical to that of preparation of zidebactam. Consequently, we have reported in the main text, the information usually reported in the caption of the schemes.
Line 650: replace “and ethyl bromo-fluoro-acetate desired.” By “and the desired ethyl bromo-fluoro-acetate.”
Done
Lines 675, 695 and 805: replace achieved by obtained
Done
Line 747: replace achieved by prepared
Done
Line 818: replace “and its group” by “and his group”
Done
Line 827: “was removed” delete “was”
Done
Line 846: replace “In this contest, in this work,” by “In the context of this work”
Done
Reviewer 4 Report
The manuscript titled "Guidelines to Synthetize Old and New β-Lactamases Inhibitors: A Review to Encourage Further Production" by Silvana Alfei and Guendalina Zuccari shows old and new β-lactamase inhibitors and a review of synthetic methods. The authors made an attempt to write a review of beta-lactamase inhibitors other than those previously published. The work presented for review is very extensive and not very clear. The content corresponding to the title proposed by the Authors starts only on page 21. The earlier text is actually an excessively elaborate introduction.
The Authors should shorten the text, limit the overall information and concentrate on information more related to synthesis and hence structure. It is necessary to supplement the references and use scientific papers.
Detailed comments:
The section entitled Introduction contains repetitions (lines 48-50 and 73-74; 33-36 and 97-100) and inaccuracies (monobactams, which have also a sulphonic acid group on the nitrogen atom of the cyclic amide, in place of the carboxylic acid one).
Fig. 1 should bear the numbering of the basic systems, and in the text of the description, the Authors should use the numbering of the compounds provided in the figure and in its caption.
Figure 2 is not quoted anywhere in the text and rightly so, it is not needed, it should be removed.
Table 1 does not contain any references. It appears to be very similar to the table found in [3], unfortunately there is no comment on this and no information on consent to use. The first column - (BLAs) * - is an incomprehensible remnant of the original, no explanation of the meaning *. Moreover, wikipedia is not the best literature source.
Line 179-183: the literature cited here [5] does not contain this information.
Line 199-218: The text only contains one reference [33] with nothing about the mechanism described here.
Table 2 - no information on where the authors obtained the presented data
The continuation of section 2 is a cluster of various redundant information, there are also abbreviations that have not been explained anywhere and will not appear again, e.g. PERs, VEBs.
Table 3: add a structure column and remove an eye-caching symbols, add information that is "Long-standing, recently approved BLEsIs, still at preclinical and clinical trials, and molecules still at the early-stage of experimentation. "
Section 4: table without references, it is incomprehensible to include columns such as „Dosage Form” and „Posology”.
Section 5: rather an overview of syntheses, often with very detailed description. Authors should add an introductory sentence to the next subsection or give up the next step of numbering e.g. lines 536-538:
5.2. Non ß-lactam DBO BLEsIs
5.2.1. Marketed DBO BLEsIs
5.2.1.1. Synthesis of Avibactam
The work also contains typographical errors, e.g. bons, contest.
Attention is also drawn to incorrect citation of literature, the names of journals are given both in full and abbreviated (with or without dots).
Author Response
Thank you for your review, please check the attachment for the authors' response.

Round 2
Reviewer 4 Report
Dear Authors,
The answers to the comments of the reviewer are satisfactory. The changes introduced by the Authors improved the quality of the article and the clarity of the information provided.